# Bayesian inference of protein structure from chemical shift data

Lars A. Bratholm[1], Anders S. Christensen[1,*], Thomas Hamelryck[2] and Jan H. Jensen[1]

[1] Department of Chemistry, University of Copenhagen, Copenhagen, Denmark
[2] Department of Biology, University of Copenhagen, Copenhagen, Denmark
[*] Current affiliation: Department of Chemistry, University of Wisconsin-Madison, Madison, WI, USA

## ABSTRACT

Protein chemical shifts are routinely used to augment molecular mechanics force fields in protein structure simulations, with weights of the chemical shift restraints determined empirically. These weights, however, might not be an optimal descriptor of a given protein structure and predictive model, and a bias is introduced which might result in incorrect structures. In the inferential structure determination framework, both the unknown structure and the disagreement between experimental and back-calculated data are formulated as a joint probability distribution, thus utilizing the full information content of the data. Here, we present the formulation of such a probability distribution where the error in chemical shift prediction is described by either a Gaussian or Cauchy distribution. The methodology is demonstrated and compared to a set of empirically weighted potentials through Markov chain Monte Carlo simulations of three small proteins (ENHD, Protein G and the SMN Tudor Domain) using the PROFASI force field and the chemical shift predictor CamShift. Using a clustering-criterion for identifying the best structure, together with the addition of a solvent exposure scoring term, the simulations suggests that sampling both the structure and the uncertainties in chemical shift prediction leads more accurate structures compared to conventional methods using empirical determined weights. The Cauchy distribution, using either sampled uncertainties or predetermined weights, did, however, result in overall better convergence to the native fold, suggesting that both types of distribution might be useful in different aspects of the protein structure prediction.

## INTRODUCTION

Protein structures can today routinely be simulated by methods such as molecular dynamics or Monte Carlo simulations, using molecular mechanics force fields (*Shaw et al., 2010*; *Karplus & McCammon, 2002*; *Snow et al., 2002*). However, this is not always a feasible method to determine a protein structure by itself. To elucidate the native protein structure efficiently, the force field energy can be augmented by restraints obtained from experiments. This immediately raises the question, how can this be done rigorously and efficiently? One pragmatic approach to this problem is to define a hybrid energy using

Corresponding author
Jan H. Jensen, jhjensen@chem.ku.dk

a penalty function, which describes the agreement between experimental data and data calculated from a proposed protein structure, together with a physical energy (such as from a molecular mechanics force field) (*Jack & Levitt, 1978*). An optimal structure in this approach could then be determined for example by minimizing the hybrid energy function

$$E_{\text{hybrid}} = w_{\text{data}} E_{\text{data}} + E_{\text{physical}}. \tag{1}$$

This approach, however, does not uniquely define neither the nature nor weight of $E_{\text{data}}$, and the resulting protein structure will depend on the choices of these.

Chemical shifts have been combined with physical energies in a multitude of ways, e.g., using weighted RMSD values or various types of harmonic constraints. Vendruscolo and co-workers implemented a 'square-well soft harmonic potential', with corresponding gradients, and were able to run a chemical shifts biased MD simulation where they successfully refined slightly denatured protein structures to a $C_{\alpha}$-RMSD of down to 0.84 Å from the corresponding crystal structures (*Robustelli et al., 2010*). The groups of Bax and Baker added the chi-square agreement between SPARTA (*Shen & Bax, 2007*) predicted chemical shift values and experimental chemical shifts with an empirical weight of 0.25 to the ROSETTA all-atom energy (*Shen et al., 2008*; *Rohl et al., 2004*). The ProCS method (*Christensen et al., 2013*) uses an approach similar to that of Bax and Baker, but with empirical weights inferred from a number of quantum mechanical calculations on representative protein models. The CHESHIRE approach (*Cavalli et al., 2007*) utilizes the experimental chemical shifts to predict secondary structure and backbone dihedral angles. These in turn are used to score molecular fragments from a database of known structures together with the chi-square agreement between the measured chemical shifts and the chemical shifts of the fragment in the database. Additionally, the final refinement phase includes a combination of physical energy terms and a term describing the correlation between experimental and back-calculated chemical shifts. A different approach was used by *Meiler & Baker (2003)*, where the contribution of the experimental chemical shifts were set relative to 1 or 0 depending on whether or not the difference to the PROSHIFT prediction (*Meiler, 2003*) exceeded a maximum tolerance. The reasoning for not using a quadratic potential was that the experimental NMR data was automatically assigned and a quadratic potential is more sensitive to assignment errors.

Clearly information from chemical shifts can be incorporated in a multitude of ways with parameters, shape and weights often tweaked by hand or estimated empirically. The inferential structure determination (ISD) principles introduced by *Rieping, Habeck & Nilges (2005)* defines a Bayesian formulation of Eq. (1), which has previously been used to determine protein structures based on NOE (*Habeck, Rieping & Nilges, 2006*; *Olsson et al., 2011*) and RDC restraints (*Habeck, Nilges & Rieping, 2008*). In the following section the equations of an ISD approach for combining the knowledge of experimental chemical shifts with a physical energy are presented.

## THEORY

In the ISD approach we seek the probability distribution of the structure $\mathbf{X}$ and a set of uncertainties $\boldsymbol{\theta}$, correlating experimental and predicted chemical shifts, given a set of experimentally measured chemical shifts $\mathbf{d}$, i.e., the probability $p(\mathbf{X}, \boldsymbol{\theta} \mid \mathbf{d})$. Using Bayes' theorem, this probability can be written as

$$p(\mathbf{X}, \boldsymbol{\theta} \mid \mathbf{d}) = \frac{p(\mathbf{d} \mid \mathbf{X}, \boldsymbol{\theta}) p(\mathbf{X}, \boldsymbol{\theta})}{p(\mathbf{d})}. \tag{2}$$

$p(\mathbf{d})$ merely serves as a normalization constant, which we need not evaluate.

We're making the basic assumption, that the deviation between predicted and experimental chemical shifts, given as

$$\Delta \delta_i = \delta_{\mathbf{X},i} - \delta_{\exp,i} \tag{3}$$

approximately follows some distribution with a variance uniquely defined by the type of nuclei ($C_\alpha$, $C_\beta$ etc.). The relevant equations for a Gaussian distribution and a Cauchy distribution (a Student's $t$-distribution with one degree of freedom), respectively, are presented in the next sections.

### Gaussian distribution

According to the principle of maximum entropy (*Jaynes, 1957*), the least informative probability distribution is the one having maximal information entropy, which given a specified mean and variance is the Gaussian distribution (*Cover & Thomas, 2012*). Assuming that each measured experimental chemical shift $\delta_{\exp,i}$ is conditional independent given the structure, the likelihood $p(\mathbf{d}|\mathbf{X}, \boldsymbol{\theta})$ is obtained as the product of the individual probabilities of all measured chemical shifts. With $i$ iterating over all $n_j$ measured chemical shifts of nuclei type $j$, this takes the form of:

$$
\begin{aligned}
p(\mathbf{d} \mid \mathbf{X}, \boldsymbol{\theta}) &= \prod_j \prod_{i=1}^{n_j} p\left(\delta_{\exp,ij} \mid \delta_{\mathbf{X},ij}, \sigma_j\right) \\
&= \prod_j \prod_{i=1}^{n_j} \frac{1}{\sigma_j \sqrt{2\pi}} \exp\left(-\frac{\Delta \delta_{ij}^2}{2\sigma_j^2}\right) \\
&= \prod_j \left(\frac{1}{\sigma_j \sqrt{2\pi}}\right)^{n_j} \exp\left(-\frac{\chi_j^2}{2\sigma_j^2}\right),
\end{aligned}
\tag{4}
$$

where $\sigma_j$ is the standard deviation in predicting chemical shifts of nuclei type $j$ and $\chi_j^2 = \sum_i^{n_j} \Delta \delta_{ij}^2$. The structure, $\mathbf{X}$, and the uncertainties in the model, $\boldsymbol{\theta}$, are assumed independent and $p(\mathbf{X}, \boldsymbol{\theta})$ can be expanded into

$$p(\mathbf{X}, \boldsymbol{\theta}) = p(\mathbf{X}) p(\boldsymbol{\theta}) = p(\mathbf{X}) \prod_j p(\sigma_j). \tag{5}$$
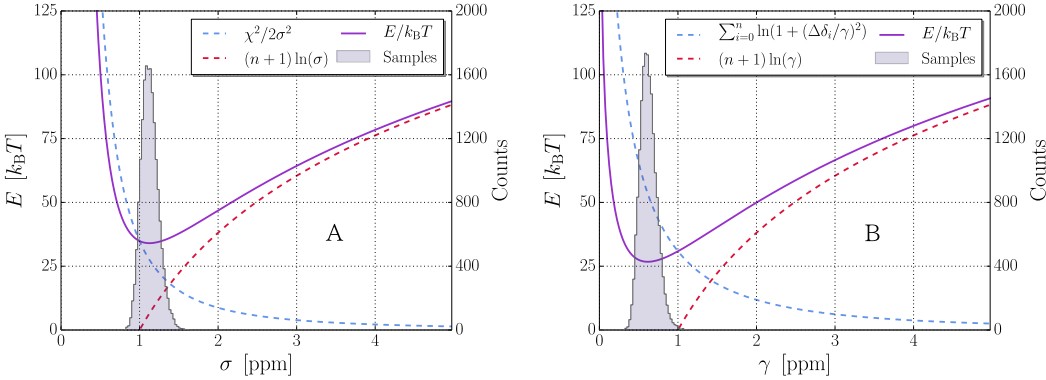

**Figure 1 Uncertainty sampling with Gaussian and Cauchy distributions.** Sampling of $\sigma$ and $\gamma$, using Jeffrey's priors, for $C_\alpha$-chemical shifts of Protein G. $n_{C_\alpha} = 54$ and $\chi^2_{C_\alpha} = 69.7$ ppm$^2$. (A) Gaussian distribution, (B) Cauchy distribution.

The prior probability for the protein structure can be expressed by the Boltzmann distribution, that is:

$$p(\mathbf{X}) = \frac{1}{Z(T)} \exp\left(-\frac{E(\mathbf{X})}{k_\mathrm{B}T}\right), \tag{6}$$

where the physical energy $E(\mathbf{X})$ could for example be approximated using a molecular mechanics force field. Note that in this case, the partition function $Z(T)$ is a normalization constant and evaluation of this is not necessary. We have little prior knowledge about $\sigma_j$ other than that it is a scale parameter. An uninformative choice of prior distribution is the Jeffreys prior (*Jeffreys, 1946*), which in this case is simply:

$$p(\sigma_j) \propto \sigma_j^{-1}. \tag{7}$$

Combining these expressions, $p(\mathbf{X}, \boldsymbol{\theta} \mid \mathbf{d})$ is thus proportional to

$$p(\mathbf{X}, \boldsymbol{\theta} \mid \mathbf{d}) \propto p(\mathbf{d} \mid \mathbf{X}, \boldsymbol{\theta}) p(\mathbf{X}) p(\boldsymbol{\theta})$$

$$\propto \prod_j \left[\sigma_j^{-n_j-1} \exp\left(-\frac{\chi_j^2}{2\sigma_j^2}\right)\right] \exp\left(-\frac{E(\mathbf{X})}{k_\mathrm{B}T}\right). \tag{8}$$

The resemblance to a hybrid energy such as in Eq. (1) is obtained by (neglecting all constant terms):

$$E_{\mathrm{hybrid}}(\mathbf{X}, \boldsymbol{\theta}) = -k_\mathrm{B}T \ln\left(p(\mathbf{X}, \boldsymbol{\theta} \mid \mathbf{d})\right)$$

$$= k_\mathrm{B}T \sum_j \left((n_j + 1) \ln(\sigma_j) + \frac{\chi_j^2}{2\sigma_j^2}\right) + E(\mathbf{X}). \tag{9}$$

From this it is seen that the standard deviations are effectively describing the weight of the experimental data. The energy dependence of $\sigma_j$ is depicted in Fig. 1.

### Conjugate prior

As discussed below, sampling uncertainties for the Gaussian model using the Jeffrey's prior leads to numerical problems. The problems arises if $\chi_j^2$ converges to zero, which leads to $\sigma_j \to 0$. This can be seen from the maximum a posteriori estimator (MAP) of $\sigma_j^2$ according to Eq. (9):

$$\sigma_{j,\text{MAP}}^2 = \frac{\chi_j^2}{n_j + 1}. \tag{10}$$

We found that these problems could be avoided by using a weakly informative prior. The conjugate prior for the variance of the Gaussian distribution $(\sigma_j^2)$, when the mean is known, can be given by an Inverse-Gamma distribution:

$$p\left(\sigma_j^2 \mid \alpha, \beta\right) = \frac{\beta^\alpha}{\Gamma(\alpha)} \left(\sigma_j^2\right)^{-\alpha-1} \exp\left(-\frac{\beta}{\sigma_j^2}\right). \tag{11}$$

$p(\mathbf{X}, \boldsymbol{\theta} \mid \mathbf{d})$ is thus proportional to

$$p(\mathbf{X}, \boldsymbol{\theta} \mid \mathbf{d}) \propto p(\mathbf{d} \mid \mathbf{X}, \boldsymbol{\theta}) p(\mathbf{X}) p(\boldsymbol{\theta})$$

$$\propto \prod_j \left[ \sigma_j^{-n_j - 2\alpha - 2} \exp\left(-\frac{2\beta + \chi_j^2}{2\sigma_j^2}\right) \right] \exp\left(-\frac{E(\mathbf{X})}{k_{\mathrm{B}} T}\right). \tag{12}$$

In contrast to Eq. (10), the maximum a posteriori estimator of $\sigma_j^2$ does not equal zero in the limit of $\chi_j^2 \to 0$ with a non-zero choice of $\beta$:

$$\sigma_{j,\text{MAP}}^2 = \frac{2\beta + \chi_j^2(\mathbf{X})}{2\alpha + 2 + n_j}. \tag{13}$$

In all the simulations where $\sigma_j$ was sampled we use Eq. (12) and $\alpha = \beta = 0.001$ (*Gelman, 2006*) unless stated otherwise.

### Marginal likelihood

Alternatively one can use the marginal likelihood where $\sigma_j$ is integrated out:

$$p(\mathbf{d} \mid \mathbf{X}) = \prod_j \int_0^\infty p\left(\mathbf{d} \mid \mathbf{X}, \sigma_j\right) p\left(\sigma_j\right) \mathrm{d}\sigma_j$$

$$\propto \prod_j \left(\chi_j^2\right)^{-\frac{n_j}{2}}. \tag{14}$$

This results in a hybrid energy of the form:

$$E_{\text{hybrid}}(\mathbf{X}) = -k_{\mathrm{B}} T \ln\left(p(\mathbf{X} \mid \mathbf{d})\right)$$

$$= k_{\mathrm{B}} T \sum_j \left(\frac{n_j}{2} \ln\left(\chi_j^2\right)\right) + E(\mathbf{X}). \tag{15}$$

## Cauchy distribution

The Cauchy and Gaussian distribution are both special cases of the Student's $t$-distribution, with degrees of freedom $\nu = 1$ and $\nu = \infty$ respectively. Compared to the Gaussian distribution, the Cauchy distribution has much heavier tails meaning that it will be less penalizing of single predictions far from the experimental values.

$p(\mathbf{d}|\mathbf{X},\boldsymbol{\theta})$ is again obtained as the product of the individual probabilities of all measured chemical shifts, with scale parameters $\gamma_j$ (equivalent to $\sigma_j$ of the Gaussian distribution):

$$p(\mathbf{d} \mid \mathbf{X}, \boldsymbol{\theta}) = \prod_j \prod_{i=1}^{n_j} p\big(\delta_{\exp,ij} \mid \delta_{\mathbf{X},ij}, \gamma_j\big)$$

$$= \prod_j \left\{ (\pi\gamma_j)^{-n_j} \prod_{i=1}^{n_j} \left[ 1 + \left(\frac{\Delta\delta_{ij}}{\gamma_j}\right)^2 \right]^{-1} \right\}. \tag{16}$$

Note that the Cauchy distribution does not reduce into an expression that depends on the $\chi_j^2$ differences (in contrast to the Gaussian). The Jeffreys prior is the same as for the Gaussian distribution:

$$p(\gamma_j) \propto \gamma_j^{-1}. \tag{17}$$

$p(\mathbf{X},\boldsymbol{\theta} \mid \mathbf{d})$ is thus proportional to

$$p(\mathbf{X}, \boldsymbol{\theta} \mid \mathbf{d}) \propto \prod_j \left\{ \gamma_j^{-(n_j+1)} \prod_{i=1}^{n_j} \left[ 1 + \left(\frac{\Delta\delta_{ij}}{\gamma_j}\right)^2 \right]^{-1} \right\} \exp\left(-\frac{E(\mathbf{X})}{k_{\mathrm{B}}T}\right). \tag{18}$$

The resemblance to a hybrid energy such as in Eq. (1) is obtained by (neglecting all constant terms):

$$E_{\mathrm{hybrid}}(\mathbf{X},\boldsymbol{\theta}) = -k_{\mathrm{B}}T\ln\big(p(\mathbf{X},\boldsymbol{\theta} \mid \mathbf{d})\big)$$

$$= k_{\mathrm{B}}T \sum_j \left\{ \left( (n_j+1)\ln(\gamma_j) + \sum_{i=1}^{n_j} \ln\left[ 1 + \left(\frac{\Delta\delta_{ij}}{\gamma_j}\right)^2 \right] \right) \right\} + E(\mathbf{X}). \tag{19}$$

# METHODOLOGY

## Computational methodology

Markov chain Monte Carlo simulations were carried out with PHAISTOS v1.0 (*Boomsma et al., 2013*) using either the multicanonical generalized ensemble via MUNINN (*Ferkinghoff-Borg, 2002*) or Metropolis–Hastings (*Metropolis et al., 1953*). Chemical shift predictions were performed with an implementation of CamShift (*Kohlhoff et al., 2009*) and the physical energy was approximated using the computational efficient PROFASI force field (*Irbäck & Mohanty, 2006*). The conformational degrees of freedom explored in the simulations were restricted to the backbone and side-chain dihedral angles ($\phi,\psi,\chi$) as well as the backbone bond angles. Backbone moves had torsion and bond

angles biased by CS-Torus (*Boomsma et al., 2014*) and Engh-Huber statistics (*Engh & Huber, 1991*) respectively, which both introduces an implicit energy. Chemical shifts were only utilized by CS-Torus for biased sampling in reference simulations where no CamShift energy term was used. The simulations were performed on AMD Opteron 2.1 GHz CPU's at ∼12 M steps/day or on Intel Xeon 3.07 GHz CPU's at ∼18 M steps/day.

### Convergence simulations

The Protein G convergence simulations were initialized from the experimental structure (PDB-id: 2OED). The simulations were run for 10 M MC steps at 300 K using Metropolis–Hastings. The physical move set was comprised of 50% local, uniform single side chain moves, 25% CRISP local moves (*Bottaro et al., 2012*) and 25% semilocal biased Gaussian step (BGS) backbone moves (*Favrin, Irbäck & Sjunnesson, 2001*).

### Structure determination simulations

The structure determination simulations were each run on 32 threads for 100 M iterations. The temperature range explored with MUNINN were set to 273 K–500 K. The physical move set was comprised of 50% local, uniform single side chain moves, 40% CRISP backbone moves and 10% backbone-DBN pivot moves (*Boomsma et al., 2008*). In the simulations where the uncertainties were dynamically adjusted, an extra 10 M Monte Carlo steps were added which sampled a change in $\sigma_j$ or $\gamma_j$ as described below. Note that these moves are essentially computationally costless, since neither chemical shifts or force field energy terms need be recomputed.

### Clustering of sampled structures

To make clustering feasible for the large amount of structures generated (320,000 structures for each combination of potential and protein), the sampled structures were converted to GIT vectors (*Røgen & Fain, 2003*) with PHAISTOS. The structures from each individual thread were subsequently divided into sets of 15 clusters with the Pleiades module of PHAISTOS (*Harder et al., 2012*) using K-means clustering (*Lloyd, 1982*). The choice of using 15 clusters is based on the suggestion of the Pleiades authors of creating 10–20 clusters. Since the clustering process is stochastic, it was performed 10 times for each thread and the optimal clustering according to the sum of squared errors were used for further analysis. From each of these clusters, a subset consisting of the 100 structures closest to the cluster centroid were selected for energy and RMSD evaluation and the median energy structures were chosen as cluster representatives. The GIT vectors can be created as output observables directly from the simulations, but in this case they were created from the simulation trajectories using the pdb2git application in PHAISTOS with the program GNU Parallel (*Tange, 2011*) used to parallelize the jobs. Re-weighting from the generalized ensemble to approximate the canonical ensemble were done automatically with Pleiades using the weighted k-means option.

## Monte Carlo move in uncertainty parameter space

The $\xi$-move which re-samples the value of the uncertainties (i.e., $\sigma$ or $\gamma$) was constructed by multiplying the previous value of $\xi$ by a sampled constant centered around 1. Detailed

balance is maintained by proposing a small change, $\xi \rightarrow \xi'$, by:

$$\xi' = \xi \cdot \exp\left(\text{rnom}\left(\sigma_\mu\right)\right), \tag{20}$$

where $\text{rnom}(\sigma_\mu)$ is a random number from a normal distribution with zero mean and standard deviation $\sigma_\mu$. A value of $\sigma_\mu = 0.1$ was found to yield a rapid and stable convergence for both the Gaussian and the Cauchy distribution.

### Issues from unexplored degrees of freedom

It was observed that CamShift predictions of $C_\beta$ chemical shifts for Isoleucine were consistently off by 3–8 ppm for the structures generated in the simulations performed with PHAISTOS. This was observed using both the CamShift implementation in PHAISTOS as well as with the standalone predictor. CamShift was trained on high quality X-ray structures where missing Hydrogens were added in accordance with the CHARMM22 topology file (*Brooks et al., 2009*). Letting the CamShift program optimize Hydrogen placement before prediction brought the accuracy of predicted Isoleucine $C_\beta$ chemical shifts in range with the prediction for the remaining amino-acids. For reference, the RMSD for $C_\beta$ chemical shift prediction of all amino-acids of a Chymotrypsin Inhibitor-II protein (CI2) structure were found to be 1.90 ppm including predictions for Isoleucine and 1.25 ppm if these predictions were excluded. As bond lengths and side-chain bond angles are not degrees of freedom in the simulations performed with PHAISTOS, the distances of $\beta$-Hydrogens and $\gamma$-Hydrogens relative to the $C_\beta$ atoms are constant. Even though this affects the Camshift prediction, it is reasonable to assume that this can be compensated to some degree by small structural perturbations. However, this distance dependence of the $C_\beta$ chemical shift prediction for Isoleucine is much larger than for the remaining amino acids (*Kohlhoff et al., 2009*) and as a result we chose to disable prediction for Isoleucine $C_\beta$ chemical shifts in the simulations.

## RESULTS AND DISCUSSION

### Problems with Gaussian weighting scheme when using a Jeffreys prior

Attempts to use predicted chemical shifts from CamShift while sampling $\sigma$ using a Gaussian model (Eq. (9)) initially proved unsuccessful. Using any structure (compact or unfolded) as starting point for the Monte Carlo simulation, it was often observed that the $\chi^2$ agreement between predicted and experimental chemical shifts would converge to zero after only a few million iterations. Naturally this leads to $\sigma \rightarrow 0$, which in turn essentially freezes the structure in the simulation, since any MC move that causes the slightest increase in chi-square will result in an enormous change in energy. If several types of chemical shifts were included in the simulation (possible chemical shift types from CamShift are $H_\alpha$, $C_\alpha$, H, N, C and $C_\beta$), the $\chi^2$ for one (random) of the included types would quickly converge to zero. One suspected reason was that the prior distribution was not well described by the more coarse grained PROFASI force field. CamShift calculations

**Table 1** Maximum likelihood estimates of $\sigma$ (or root-mean-square deviation (RMSD)) obtained from the CamShift training set, compared to means extracted from a $10^7$ MC step simulation using the Gaussian model (see text). Shown values are in units of ppm.

| | $C_\alpha$ | $H_\alpha$ | N | H | C | $C_\beta$ |
|---|---|---|---|---|---|---|
| CamShift training set | 1.22 | 0.26 | 2.78 | 0.56 | 1.12 | 1.19 |
| Frozen simulation[a] | 1.13 | 0.26 | 3.53 | 0.52 | 1.06 | 1.21 |
| Free simulation[a] | 1.03 | 0.20 | 2.92 | 0.46 | 1.16 | 1.23 |

**Notes.**
[a] Estimated over the last $10^6$ MC steps.

**Table 2** Maximum likelihood estimates of $\gamma$ obtained from the CamShift training set, compared to means extracted from a $10^7$ MC step simulation using the Cauchy model (see text). Shown values are in units of ppm.

| | $C_\alpha$ | $H_\alpha$ | N | H | C | $C_\beta$ |
|---|---|---|---|---|---|---|
| CamShift training set | 0.70 | 0.19 | 1.87 | 0.31 | 0.74 | 0.77 |
| Frozen simulation[a] | 0.62 | 0.17 | 1.90 | 0.32 | 0.64 | 0.69 |
| Free simulation[a] | 0.43 | 0.05 | 1.57 | 0.25 | 0.67 | 0.55 |

**Notes.**
[a] Estimated over the last $10^6$ MC steps.

were therefore redone using the OPLS-AA/L force field (*Kaminski & Friesner, 2001*). This, however, led to identical results.

CamShift (and most likely other similar predictors) is able to make relatively large changes in prediction, from a small perturbation in the structure. Combined with sampling of $\sigma$, this can drive the simulation into an energy minimum with essentially zero error in the chemical shift prediction, even though the structure may or may not be anything like the native structure. We found the Cauchy distribution to be less sensitive to divergence of the scale parameter and to perform better as an uninformative model in our case. As an alternative to the Jeffreys prior, a weakly informative conjugate prior for the Gaussian model did not show these sampling issues.

## Convergence of scale parameters

The convergence of the scale parameters for the Gaussian and Cauchy distributions ($\sigma$ and $\gamma$ respectively), with chemical shifts predictions by CamShift (*Kohlhoff et al., 2009*), were explored by starting a simulation with PHAISTOS (*Boomsma et al., 2013*) from the native structure of Protein G (PDB:2OED (*Ulmer et al., 2003*)). Experimental chemical shifts were obtained from Ref-DB (*Zhang, Neal & Wishart, 2003*) (RefDB:2575 (*Orban, Alexander & Bryan, 1992*)). For each model, a $10^7$ MC step simulation was performed: keeping the structure fixed, only sampling uncertainties (frozen), and a simulation where the atomic coordinates (**X**) was sampled as well (free). Tables 1 and 2 shows the mean of the sampled parameters from the last $10^6$ steps together with the maximum likelihood values obtained from the CamShift training set for reference.

Using a Gaussian distribution, the parameters in the 'frozen' simulation all converged within 0.1 ppm to the reported values from the CamShift training set, with the exception of the N nuclei which deviated by 0.75 ppm. The RMSDs presented in Table 1 for the CamShift training set were based on predictions on 7 proteins and, using a larger data set of 28 proteins, the average RMSD for the N nucleus increased from 2.78 ppm to 3.01 ppm (*Kohlhoff et al., 2009*). Thus, the slightly higher mean for N seems reasonable. Allowing the structure and weight parameters to be sampled simultaneously in the 'free' simulation overall lowered the RMSD of the prediction as expected, since the accepted structures in the Monte Carlo simulation will be biased by the correlation of predicted and experimental chemical shifts. However, the RMSD increased moderately for the C nucleus and slightly for $C_\beta$, indicating that the chemical shift prediction of C and $C_\beta$ are less sensitive to changes in local structure than the four other nuclei.

In the simulations using a Cauchy distribution, the 'frozen' values were seen to be similar to the CamShift data set (within 0.1 ppm). When physical moves were introduced in the 'free' simulation, the sampled parameters were again found to be lowered, but remained within 0.3 ppm. Surprisingly, $\gamma$ for $H_\alpha$ went from 0.17 ppm to 0.05 ppm, with similar values found when repeating the simulation. The $\chi^2$ error in the prediction of $H_\alpha$ chemical shifts were similar to that obtained with the Gaussian potential, indicating that the error in prediction for $H_\alpha$ atoms had several outliers. Since the Cauchy distribution is less sensitive to outlier values, these will have a lesser effect on the sampled parameters than for the Gaussian.

## Comparison of weighting schemes in structure determination

A series of simulations starting from an unfolded state were performed on ENHD (PDB: 1ENH (*Clarke et al., 1994*), BMRB:15536 (*Religa, 2008*)), Protein G and the SMN Tudor Domain (PDB: 1MHN (*Sprangers et al., 2003*), RefDB:4899 (*Selenko et al., 2001*)) to compare how different weighting schemes performed for structure determination. The probabilistic schemes used included three Gaussian models: one using the maximum likelihood estimates of $\sigma$ from the CamShift training set (Gaussian/fixed); one where the values of $\sigma$ were sampled (Gaussian/sampled); and one using the marginalized distribution (Gaussian/marginalized). Similarly, two Cauchy models were tested: one using maximum likelihood values for $\gamma$ from the CamShift training set (Cauchy/fixed), and one where the values for $\gamma$ were sampled (Cauchy/sampled). As reference, the square well potential of *Robustelli et al. (2010)*, which was made specifically for refinement with the CamShift model, were included in the simulations with different weights (Square well/$\alpha = 1$, Square well/$\alpha = 5$).

In all simulations, the generative predictive model CS-Torus (*Boomsma et al., 2014*) was used to sample backbone dihedral angles from a distribution biased by the amino-acid sequence. Chemical shifts can provide local information to the CS-Torus model to further improve the biased sampling, but this was not utilized in any simulations using CamShift predictions. Although including chemical shifts in the sampling would most likely improve the simulation results, we chose to keep the CamShift energy terms as the only bias from

**Table 3 Different weighting schemes used in the protein folding simulations.** In the columns to the left, the number of threads, out of a total of 32, sampling structures below 2 and 4 Å $C_\alpha$-RMSD respectively to the reference structure is shown. The sampled structures from each thread were divided into clusters and representative structures for each cluster were selected as the structure median in PROFASI+CamShift energy, from the 100 structures closest to the cluster centroid. The $C_\alpha$-RMSD in Å of the lowest-energy cluster representative is shown below in the columns to the right.

| | Threads (out of 32) sampling below 2Å (left) and 4Å (right) | | | | | | Lowest-energy RMSD (Å) | | |
| --- | --- | --- | --- | --- | --- | --- | --- | --- | --- |
| | ENHD | | Protein G | | SMN | | ENHD | Protein G | SMN |
| Gaussian/fixed | 32 | 32 | 0 | 7 | 29 | 30 | 3.67 | 3.11 | 3.11 |
| Gaussian/sampled | 32 | 32 | 4 | 15 | 13 | 20 | 2.15 | 3.03 | 5.88 |
| Gaussian/marginalized | 32 | 32 | 1 | 16 | 7 | 14 | 4.24 | 2.72 | 6.06 |
| Cauchy/fixed | 32 | 32 | 9 | 25 | 15 | 21 | 1.94 | 1.15 | 2.58 |
| Cauchy/sampled | 32 | 32 | 13 | 24 | 11 | 16 | 1.87 | 2.82 | 5.51 |
| Square well/$\alpha = 1$[a] | 19 | 22 | 2 | 12 | 14 | 18 | 2.29 | 3.14 | 3.71 |
| Square well/$\alpha = 5$[a] | 32 | 32 | 0 | 1 | 1 | 5 | 3.82 | 5.83 | 1.91 |
| CS-Torus[b] | 4 | 27 | 8 | 25 | 0 | 0 | 19.2 | 3.01 | 8.33 |

**Notes.**

[a] Weights, $\alpha$, of 1 and 5 were used by Robustelli et al.

[b] Lowest-energy cluster representatives for the CS-Torus simulations were selected from PROFASI energy alone.

the experimental chemical shifts. To display the effect of using a non-local chemical shift predictor like CamShift instead of relying on local information alone in the sampling, simulations using chemical shifts in the CS-Torus model, rather than with CamShift prediction, were run as well.

Thirty-two folding simulations were run for each potential and protein for 100 M MC steps using the PROFASI (*Irbäck & Mohanty, 2006*) force field and a CamShift energy term. For each set of simulations, the sampled structures from each thread were subsequently split into clusters as described in the Methodology section, and cluster representatives were selected as the structures median in energy, from the 100 structures closest to the cluster centroid. Table 3 shows the number of threads sampling structures below 2 and 4 Å $C_\alpha$-RMSDs to the native structures as well as the RMSDs for the cluster representative with the lowest PROFASI+ CamShift energy. The residue ranges used to calculate the RMSDs were 5–54 for ENHD, all residues for Protein G and 4–56 for the SMN Tudor Domain.

### *Convergence of sampling*

The data in Table 3 shows that for certain potentials and proteins, several threads failed to sample near-native structures. For ENHD all potentials but the CS-Torus model and square well/$\alpha = 1$ potential sampled structures below 2 Å $C_\alpha$-RMSD for all threads. While more than 20 threads sampled structures below 4 Å for both the CS-Torus and square well model, only 4 threads sampled structures below 2 Å for CS-Torus. For Protein G no threads for the Gaussian/fixed and square well/$\alpha = 5$ potentials sampled structures below 2 Å. The square well/$\alpha = 1$, Gaussian/marginalized and Gaussian/sampled potentials only sampled
these near-native states with a few threads, while the Cauchy potentials and the CS-Torus model showed the fewest sampling issues.

Looking closer at the threads never sampling structures close to native for Protein G, it is found that the majority of these never progressed past a local energy-minimum with an alternative conformation where two $\beta$-strands have interchanged position (Fig. 2). Taking the median structure of the most dense cluster as representative for each thread, 27 of these show this incorrect fold for the Gaussian/fixed potential and 26 for the square well/$\alpha = 1$ potential. The Cauchy distributions shows the opposite trend with 25 correct folds for both potentials, while the structures from the Gaussian/sampled and Gaussian/marginalized simulations had 14 and 11 correctly folded, respectively. For all of these potentials, the densest clusters of each thread have either this misfold or the correct structure. While the square well/$\alpha = 5$ potential seem to find completely incorrect structures, the CS-Torus simulations finds the correct overall fold in 20 threads. The remaining CS-Torus threads are partly unfolded and none of them have the misfolded structure found in the simulations with CamShift energy terms. Finally, for the SMN Tudor Domain, the Gaussian/fixed model sampled structures below 2 Å for nearly all threads. The CS-Torus model and square well/$\alpha = 5$ potential for 0 and 1 thread(s) respectively, while the remaining potentials sampled below 2 Å for around a third of the threads.

Ideally, the simulations with a given potential samples structures close to native consistently well for all proteins, which was not the case for the Gaussian/fixed model, square well/$\alpha = 5$ potential, the CS-Torus reference model and to a lesser extent the Gaussian/sampled model. The two Cauchy potentials was most likely to sample low-RMSD structures across the three proteins. Due to limitations of the MUNINN implementation in PHAISTOS at the time the simulations were run, the multicanonical generalized ensembles from each thread cannot be re-weighted to approximate a single canonical ensemble, and clustering of structures must be done on a per-thread basis. Since cluster densities can't readily be compared across threads, the structure clusters are evaluated from the force field and CamShift energy.

### *Lowest-energy clusters*

Table 3 shows for each potential and protein the $C_\alpha$-RMSDs to native for the lowest-energy structures found by clustering. There is no clear consensus of which potentials result in the most accurate structures overall based on the RMSD values. Visually (Figs. S1–S6) all but CS-Torus has the correct fold for ENHD, with the Gaussian/fixed, Gaussian/marginalized and square well/$\alpha = 5$ structures being less compact than the crystal structure. For protein G, only the square well/$\alpha = 5$ potential shows a slight misfold, and the overall somewhat high RMSDs is again due to slightly less compact structures, as well as a small displacement of beta-sheet positions for all but the CS-Torus and Cauchy/fixed models. Although the misfold shown in Fig. 2 was prevalent in the simulations in many threads, none of the lowest-energy structures have these interchanged $\beta$-strand positions. For the SMN Tudor Domain, the difference in RMSDs between the potentials is mainly due to the protein tails not being correctly placed in a compact structure.

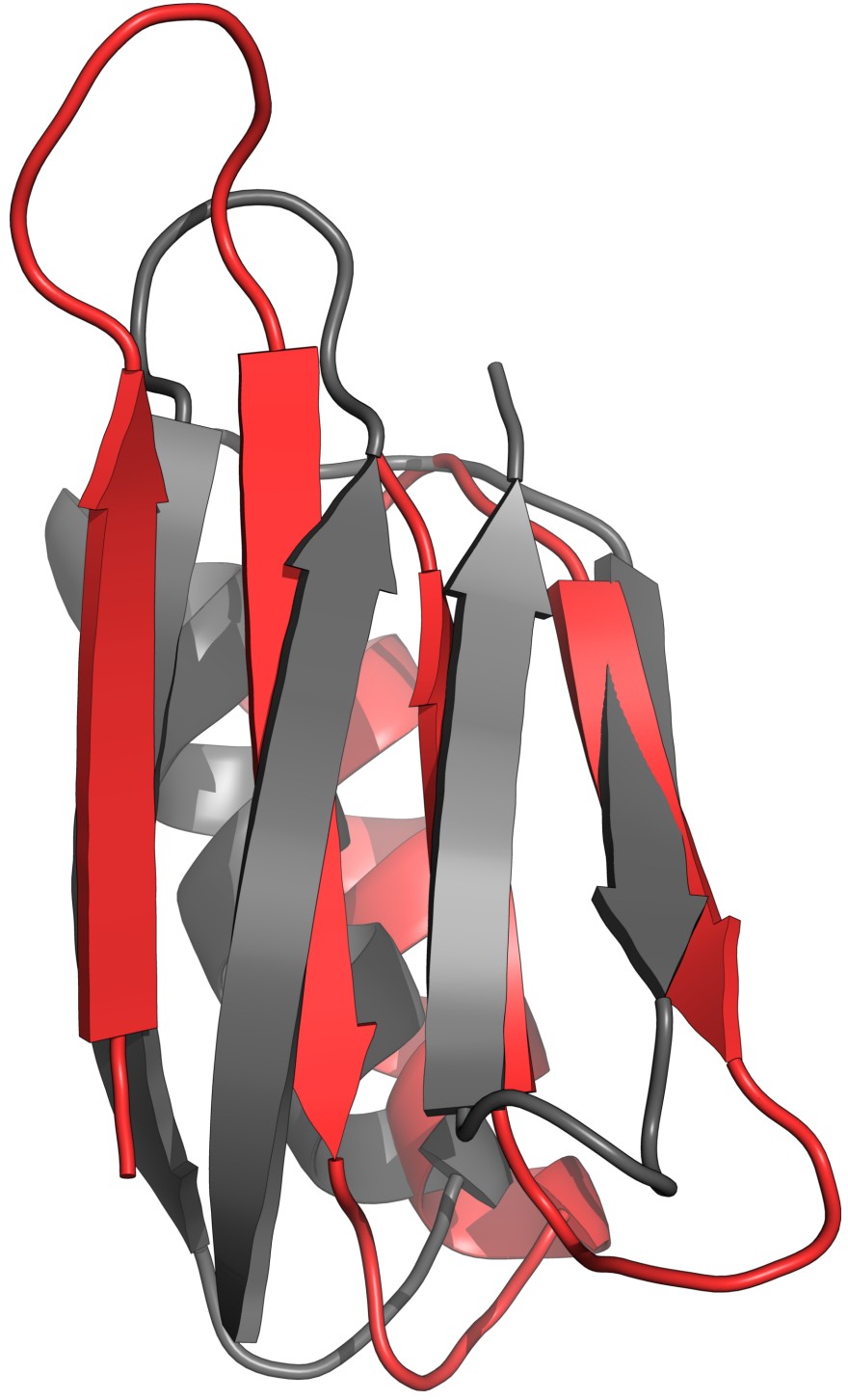

**Figure 2 Local energy-minimum of Protein-G.** Crystal structure (grey) and local energy-minimum conformation (red) of Protein G. Figure made with PyMOL (*Schrödinger LLC, 2010*).

**Table 4  $C_\alpha$-RMSDs in Å of the lowest-energy cluster representative, when a solvent exposure energy term (HSEMM) is added to re-score the structures.**

| | Lowest-re-scored-energy RMSD | | |
| --- | --- | --- | --- |
| | ENHD | Protein G | SMN |
| Gaussian/fixed | 1.40 | 2.45 | 2.23 |
| Gaussian/sampled | 1.03 | 1.29 | 1.24 |
| Gaussian/marginalized | 1.11 | 1.00 | 3.81 |
| Cauchy/fixed | 1.40 | 1.16 | 1.55 |
| Cauchy/sampled | 1.86 | 0.86 | 2.50 |
| Square well potential/$\alpha = 1$[a] | 1.15 | 1.37 | 3.05 |
| Square well potential/$\alpha = 5$[a] | 0.96 | 4.35 | 1.91 |
| CS-Torus[b] | 3.88 | 1.57 | 9.18 |

**Notes.**

[a] Weights, $\alpha$, of 1 and 5 were used by Robustelli et al.

[b] Lowest-energy cluster representatives for the CS-Torus simulations were selected from PROFASI+HSEMM energy alone.

As mentioned above, the obtained structures from the lowest-energy clusters are in general less compact than the crystal structures. This is a result of additional compactness terms being excluded in the simulations such that the effect of using different potentials for modelling the discrepancy between observed and predicted chemical shifts might be more clear. In nearly all of the simulations higher energy clusters exists that have lower RMSDs to the native structure, suggesting that near-native structures are sampled, but the compactness of the protein isn't properly described by the force field. Evaluating sampled structures with energy terms not included in the Monte Carlo simulations is problematic, since the energy can fluctuate greatly with small changes in local structure. However when entire clusters of structures are evaluated this becomes less of a problem, especially when coarse grained energy terms is used in addition to the energies obtained from the simulations. The half-sphere exposure mixture model (HSEMM), implemented in PHAISTOS for modelling solvent exposure, is a variation of the multibody multinomial model (MuMu) (*Johansson & Hamelryck, 2013*) with the environment of residue $i$ described by four features: the secondary structure according to CS-Torus, the backbone hydrogen bond network and the half sphere exposure up and down measure (*Hamelryck, 2005*). For every cluster, the energy from HSEMM was calculated and added to the total energy of the structures, with the hydrogen bond network feature integrated out to enforce the coarse grained characteristics of the model.

The results are summarized in Table 4 and show that the lowest-energy clusters re-scored with the solvent exposure term all have lower or similar RMSDs to the clusters evaluated with just the PROFASI+ CamShift energies. Sampling of the uncertainty when using the Gaussian distribution results in the structures closest to native, with RMSDs below 1.5 Å for all three proteins. For the Cauchy distribution, sampling the uncertainties does not seem to be an improvement over using predetermined weights, but both

approaches gives better structures overall than the remaining potentials. Furthermore, it is clear that the non-local information provided by the CamShift model greatly improves structure sampling, as shown by the relatively poor performance of the simulations using only CS-Torus.

## CONCLUSION

We present a probabilistic method for biasing protein structure simulations with experimentally measured chemical shifts, based on the inferential structure determination formalism (ISD) (*Rieping, Habeck & Nilges, 2005*). In this formalism, the weighting of experimental data can be determined entirely by the data itself, the predictive model and the physical force field.

Simulations were performed on three small proteins (ENHD, Protein G and SMN Tudor Domain) for a Gaussian and Cauchy-based probability distribution, using the chemical shift predictor CamShift (*Kohlhoff et al., 2009*). The ISD-determined uncertainties were found to correspond well to the empirically determined uncertainties in the CamShift predictions. Furthermore sampling the uncertainties as part of the protein structure determination simulations, lead to improved accuracy of the predicted structures when a Gaussian potential was used. Using a Cauchy potential with either sampled or fixed uncertainties did, however, show overall better convergence to the native fold, suggesting that the simulations are less likely to get stuck in local minima with these potentials. Additionally, the importance of capturing non-local information from experimental chemical shifts have been shown by comparing the use of the CamShift predictor to the local-only CS-Torus model.

### Funding

Anders S. Christensen is funded by the Novo Nordisk STAR program. Lars A. Bratholm is funded by the Lundbeck Foundation. Thomas Hamelryck was supported by the University of Copenhagen 2016 Excellence Programme for Interdisciplinary Research (UCPH2016-DSIN). The funders had no role in study design, data collection and analysis, decision to publish, or preparation of the manuscript.

### Grant Disclosures

The following grant information was disclosed by the authors:
Novo Nordisk STAR program.
Lundbeck Foundation.
University of Copenhagen 2016 Excellence Programme for Interdisciplinary Research:
UCPH2016-DSIN.

### Competing Interests

The authors declare there are no competing interests.

## Author Contributions

- Lars A. Bratholm and Anders S. Christensen conceived and designed the experiments, performed the experiments, analyzed the data, contributed reagents/materials/analysis tools, wrote the paper, prepared figures and/or tables, reviewed drafts of the paper.
- Thomas Hamelryck conceived and designed the experiments, contributed reagents/materials/analysis tools, reviewed drafts of the paper.
- Jan H. Jensen conceived and designed the experiments, contributed reagents/materials/analysis tools, wrote the paper, reviewed drafts of the paper.

## Supplemental Information

Supplemental information for this article can be found online at http://dx.doi.org/10.7717/peerj.861#supplemental-information.

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
