# Peer review of "Bayesian inference of protein structure from chemical shift data"

_PeerJ, doi:10.7717/peerj.861_

## Round 0.1 · original submission · Major Revisions

Please address the concern raised by the second reviewer about the singular behavior of the Jeffrey prior.

Reviewer 1 ·

Basic reporting

One sections, THEORY, does not conform to standard PeerJ manuscript style. However, it helps the overall presentation of the work and is therefore warranted, in my opinion.

Experimental design

No comments

Validity of the findings

No comments

Additional comments

The ms explores various ways of incorporating experimental chemical shift data with physical molecular mechanics modeling, in order to enhance protein structure prediction methods. In particular, a statistical framework is developed for carrying out the incorporation of data into a hybrid potential energy function, which can then be minimized. An interesting result is that there does not seem to be a single optimal procedure but that it depends on the protein. In other words, it may be difficult to a priori select the best procedure in a true blind prediction situation. The data obtained are robust and statistically sound, and the conclusions reached are appropriately stated and well-supported. As such, the work is a nice addition to the field and will no doubt aid future work in the area.

Reviewer 2 ·

Basic reporting

1. The introduction (and partially the abstract) is inaccurate and should be rewritten.
So, just as an example, the authors miss that CHESHIRE uses chemical shifts not only to predict torsion angles but also in the refinement phase where the energy function contains a contribution that depends on the correlation between experimental and calculated shifts.

2. The theory section contains errors. So equation (7) and (17) are wrong... because the Jeffrey prior in not normalizable.

3. "use of the Jeffrey’s prior and the Gaussian model with the empirical
chemical shift predictor CamShift leads to numerical problems. The problems arises if χ2j converges to zero, which leads to σ j → 0. "
And this means that Jeffrys prior should not be used ... because we can not "blame" a predictor if the Chi2 is zero. A perfect predictor has to have Chi2=0 for the right structure or set of structures (if the correspondence is not 1-1).

4.Issues with CamShift...and Isoleucine, but it turns out to be a problem with PHISTOS hydrogen placement....

Experimental design

The authors should rethink the structure of the paper and state clearly what are the objectives and the logic they follow.

An example.

They use a Jeffry prior for sigma^2 (and a Gaussian error model) so that the "system" does not have any free parameters. Fine. But this leads to a singular probability distribution (E_hyb = -infty if Chi2=0, regardless of E_phys). So they regularise it with a semi-informative inverse-gamma prior. Fine again. The problem is that this regulariser has 2 free parameters alpha and beta (instead of 1: sigma^2) that they set arbitrarily to 0.001. So what is the point of the exercise? Going from a single parameter that has a "meaning": the variance of the prediction error on a test set, to 2 parameters that are arbitrary set to 0.001?

An other example...
E_hyb = -infinity if chi^2=0.
Every meaningful model should have an energy function E>c over the configuration space otherwise the system will find a way to go there!

Validity of the findings

No comments

---

## Round 0.2 · accepted · Accept

Both reviewers are now satisfied with the present version, and I think that the paper is a relevant contribution to the field.

Reviewer 1 ·

Basic reporting

No comments

Experimental design

No comments

Validity of the findings

No comments

Additional comments

No comments

Reviewer 2 ·

Basic reporting

The manuscript is acceptable in current form.

Experimental design

No additional comments.

Validity of the findings

No additional comments.